# Growth Inhibitory and Selective Pressure Effects of Sodium Diacetate on the Spoilage Microbiota of Frankfurters Stored at 4 °C and 12 °C in Vacuum

**DOI:** 10.3390/foods10010074

**Published:** 2021-01-01

**Authors:** John Samelis, Athanasia Kakouri

**Affiliations:** Hellenic Agricultural Organization ‘DIMITRA’, Dairy Research Department, 45221 Katsikas, Ioannina, Greece; kakouriathanasia@yahoo.gr

**Keywords:** frankfurters, sodium diacetate, lactate, *Lactobacillus sakei*, *Leuconostoc carnosum*, *Leuconostoc mesenteroides*

## Abstract

This study evaluated microbial growth in commercial frankfurters formulated with 1.8% sodium lactate (SL) singly or combined with 0.25% sodium diacetate (SDA), vacuum-packaged (VP) and stored at 4 °C and 12 °C. Standard frankfurters without SDA, containing 0.15% SL, served as controls (CN). Lactic acid bacteria (LAB) were the exclusive spoilers in all treatments at both storage temperatures. However, compared to the CN and SL treatments, SL + SDA delayed growth of LAB by an average of 5.1 and 3.1 log units, and 3.0 and 2.0 log units, respectively, after 30 and 60 days at 4 °C. On day 90, the SL + SDA frankfurters were unspoiled whereas the SL and CN frankfurters had spoiled on day 60 and day 30 to 60, respectively. At 12 °C, LAB growth was similar in all treatments after day 15, but strong defects developed in the CN and SL frankfurters only. Differential spoilage patterns were associated with a major reversal of the LAB biota from gas- and slime-producing *Leuconostoc mesenteroides* and *Leuconostoc carnosum* in the CN and SL frankfurters to *Lactobacillus sakei/curvatus* in the SL + SDA frankfurters. Thus, SL + SDA extends the retail shelf life of VP frankfurters by delaying total LAB growth and selecting for lactobacilli with a milder cured meat spoilage potential than leuconostocs, particularly under refrigeration.

## 1. Introduction

Feasible hygienic measures adopted during peeling, slicing and packaging of ready-to-eat (RTE) cooked cured meat products in commercial meat processing plants cannot guarantee complete avoidance of post-process contamination with various spoilage microorganisms [1,2]. Certain lactic acid bacteria (LAB) species, or even groups of closely related LAB species specifically adapted to meat, are the primary to exclusive microbial spoilers of refrigerated, vacuum-packaged (VP) or modified-atmosphere-packaged (MAP) cooked meat products by producing in-package acid, gas, ropy slime and acidic off-odors during storage [3,4,5,6]. Hence, managing LAB spoilage in the meat processing industry remains a major challenge [1,2].

During the last two decades, there has been an increasing use of combinations of organic acid salts, mainly sodium lactate (SL) or potassium lactate (PL) with sodium acetate (SA) or sodium diacetate (SDA), for controlling post-thermal *Listeria monocytogenes* contamination and growth in cold-stored VP or MAP frankfurters, sliced ham and other cooked cured meat products [7,8,9,10,11,12,13]. In specific, 2–4% SL or preferably PL, singly or combined with up to 0.25% SA or SDA, are currently permitted and applied as commercial antilisterial blends in the meat processing industry worldwide [14,15]. Because meat spoilage LAB and other non-LAB bacteria are also inhibited by organic acids or their salts in laboratory media in vitro [4,16,17,18,19], inclusion in the formulation of SL/PL with SA or SDA as antilisterial agents extends the shelf life of cooked meat products in addition to increasing their safety [1,12,14,15]. Several previous challenge studies focused on the antilisterial effects of SL/PL, SA, SDA and other organic acid salt treatments during storage of various processed meat products have reported parallel growth retardation or prolonged inhibitory effects on the spoilage microbiota which, however, were based on total bacterial and LAB enumerations only [7,8,9,10,12].

Published studies that specifically address selective effects of lactates, acetates and other organic acid salts during storage of commercial RTE cooked cured meat products are scarce and lack comparative identification data regarding the main LAB species prevailing naturally within treatments [12,19]. Additionally, with the profound exception of two recent studies on fresh (raw) pork sausage [20,21], there is a lack of comparative research-based data regarding differential selective pressure effects caused in situ by commercial blends of lactate and acetate salts against meat spoilage bacteria, particularly in cooked meat products; although it has been well documented since the mid 80 s that several spoilage LAB specific to meat, such as *Carnobacterium* spp. and certain *Weissella* or *Leuconostoc* spp., are more acid-sensitive, particularly to acetate, than the *Lactobacillus sakei/curvatus* group in vitro [17,18,22,23].

In summary, intrinsic selective pressure effects of lactate and acetate salts on the growth dynamics of LAB in cooked meats remain largely unexplored despite the great progress in mapping the meat spoilage ecology overall by culture-dependent molecular identification tools and next generation sequencing approaches in recent years [2,24,25,26]. Usually, identified LAB isolates have been collected from random and quite diverse cooked meat product types purchased from national markets [5,27,28,29,30]. Relatively fewer studies have identified the dominant LAB biota at different processing steps of industrial cured meat products, mostly cooked ham, to evaluate technology-induced selections towards the spoilage microbial associations [31,32,33,34]. However, neither the latter studies have addressed selective pressure effects of organic acid salts in ham formulations, including a recent large and very dynamic microbiota survey across an industrial processing line of French cooked hams [34].

Meanwhile, the need to conduct factory-specific studies to evaluate whether lactate and acetate salts may alter the LAB ecology in ways to delay, modify or reduce spoilage defects of cooked cured meat products was first pointed out in Greece by Samelis and Georgiadou [35]. A later preliminary industrial study on VP Greek frankfurters formulated with 1.8% (3% of a 60% *w/w* commercial product) SL, or 1.8% SL plus 0.25% SA, showed that SL + SA selected for *Lb. sakei/curvatus* strains with a mild spoilage potential, whereas the SL frankfurters spoiled more offensively due to high prevalence of *Leuconostoc* spp. that formed excessive in-package gas and ropy slime during storage at 4 °C and mainly at 12 °C [36]. However, LAB growth was faster in the SL + SA than the SL frankfurters during storage at 4 °C, probably because SA at 0.25% favored the growth of acetate-resistant *Lb. sakei/curvatus* strains [36]. Therefore, this study aimed to improve control of the total bacterial growth, as well as, to address potential selective pressure effects on the spoilage LAB species association by the inclusion of 1.8% SL singly or combined with 0.25% SDA in replacement of 0.25% SA, in industrially formulated VP frankfurters stored under refrigeration (4 °C) or thermal abuse (12 °C) conditions.

## 2. Materials and Methods

### 2.1. Formulation, Preparation, Storage and Sampling of Industrial Frankfurter Samples

Frankfurters (12 cm length; 24 mm diameter) were prepared in a local industrial meat processing plant (BI.K.H. s.a., Filippiada, Epirus, Greece) from pork meat and fat under standard commercial manufacturing conditions. The basic (standard) frankfurter formula contained 60% pork meat. Emulsified collagen comprised 11% of the final batter. For collagen emulsification, 2.27 kg of SL (60% *w/w* commercial product; Purac Inc. Lincolnshire, IL, USA) were used for treating 100 kg of pork skin, accounting for 0.15% pure SL already in the basic batter; thus, 0.15% SL was subtracted from the SL added to treatments below. Further details on the frankfurter recipe (% composition) were not provided because they were considered a confidential ”know-how” proprietary to the company. However, necessary precautions were taken during preparation of the experimental batters: (i) the production manager guaranteed that the standard frankfurter formula did not contain other antimicrobials, except common cured meat ingredients, sodium chloride, dextrose, powdered milk, starch, spices, nitrites, phosphates, Na-erythrobate, Na-glutamate and Na-ascorbate; (ii) all curing ingredients were added to the cutter in our presence; (iii) the desired SL and SDA concentrations were weighed and added to the frankfurter batters by the production manager under our close supervision.

The preparation, storage and sampling of frankfurters are summarized in Figure 1.

Briefly, the basic frankfurter batter was prepared and divided in 150-kg portions, which were mixed separately in the cutter to incorporate the salts. Treatments included addition of 1.8% SL (3% of the aforementioned 60% *w/w* Purac product) and 1.8% SL plus 0.25% SDA (Carlo Erba, sodium hydrogen di-acetate FCC, code 651472; Carlo Erba Reagents, Val de Reuil, France). Standard frankfurters that contained 0.15% pure SL, manufactured without SDA and/or additional SL, served as controls (CN). Preparation of the CN samples was preceded followed by the preparation of the SL and SL + SDA samples to avoid cross-contamination of the batters and washing the cutter bowl between treatments. Following cooking, cooling, overnight storage at 4 °C and removal of the product casings, the peeled frankfurters of each treatment were vacuum packaged (VP; 8 links/pack) in the plant, transported to our microbiology laboratory in insulated polystyrene iceboxes, and distributed in two cooling incubators (Velp Scientifica FOC 225I, Usmate, Milano, Italy), at 4.0 ± 0.1 °C and 12.0 ± 0.1 °C. Two independent frankfurter experimental trials, processed on different production days, were studied. For each experimental trial, two random VP frankfurter sample packs of each treatment were analyzed microbiologically and for pH and overall appearance on days 0, 7, 15, 30, 60, and 90 of storage. Additionally, two random, freshly-VP frankfurter samples per treatment were analyzed for moisture, fat, protein and salt contents in the industrial plant’s laboratory, where the treatments were also compared organoleptically, as described below.

### 2.2. Microbiological Analyses

Frankfurter samples (25 g) were transferred aseptically to 225 mL of 0.1% (*w/v*) buffered peptone water (BPW) (Neogen Media; formerly Lab M, Heywood, UK), and homogenized in a stomacher (Lab Blender, Seward, London, UK) for 1 min at low speed and 1 min at high speed at room temperature. Serial decimal dilutions in 0.1% BPW were prepared and 1 mL or 0.1 mL samples of the appropriate dilutions were poured or spread, in duplicate, on total and selective agar plates. All samples were analyzed for total spoilage microbiota enumerated on tryptone soy agar with 0.6% yeast extract (TSAYE) (Lab M), incubated at 30 °C for 72 h, and for total LAB enumerated on de Man, Rogosa & Sharpe (MRS) agar (Lab M), incubated at 25 °C for 72 h. Anaerobic incubation of the MRS plates was avoided to facilitate growth of all spoilage LAB types at 25 °C and the comparison with the total mesophilic (TSAYE) bacterial counts at 30 °C. Additionally, all frankfurter treatments were analyzed on days 0, 15, 30 and 90 of storage for the following microbiota expected to be subdominant or absent: enterococci on kanamycin aesculin azide agar (Lab M), incubated at 37 °C for 48 h; total staphylococci on Baird-Parker agar base (Lab M) with egg yolk tellurite (supplement X085; Lab M), incubated at 37 °C for 48 h; total enterobacteria in violet red bile glucose agar (Merck, Darmstadt, Germany), overlayed with 5 mL of melted (45 °C) same medium and incubated at 37 °C for 24 h; pseudomonad-like and other psychrotrophic gram-negative bacteria on *Pseudomonas* agar base (Lab M) with cephalothin-fucidin-cetrimide (CFC; supplement X108; Lab M), incubated at 25 °C for 48 h; sulfite-reducing clostridia in 50-mL tubes containing 20 mL of melted sulfite-polymyxin-sulfadiazine agar (SPS, Merck), inoculated with 10 mL from the first dilution (equal to 1 g of meat), cooled rapidly in ice-water, overlayed with 2 mL of sterile paraffin oil to exclude oxygen, and incubated at 37 °C for 24 h; and, yeasts on rose bengal chloramphenicol agar (Merck), incubated at 25 °C for 5 days. The microbial quantification methods and rapid confirmatory tests of the colonies grown on the above selective media were conducted according to the procedures described by Samelis et al. [37].

### 2.3. Physicochemical Analyses

The pH of all frankfurter samples during storage at both temperatures was measured in 5-g sample portions homogenized with 45 mL of distilled water. A Jenway 3510 digital pH meter (Essex, UK) equipped with a glass electrode was used for the measurement. The moisture, fat, protein and salt contents of the frankfurter samples were determined before storage (day 0), according to the AOAC official methods [38].

### 2.4. Sensory Evaluation

No formal sensory panel testing with scale score sheets was conducted for the purposes of this study. However, the frankfurter treatments (CN, SL, SL + SDA) of each production trial were evaluated comparatively by five experienced members of the industrial personnel, who judged whether addition of SL and SL + SDA in the batter might have affected the appearance, odor-flavor and taste of the fresh (day-0) frankfurters, compared to the CN frankfurters. For taste evaluation, the peeled frankfurters were immersed in an 80 °C water-containing cooking jar for 5 min, cooled, tested and the judgments of the industrial panelists were recorded and compared. Next, during storage, all VP frankfurter samples were checked for accumulation of milky exudates, gas and/or slime in-package and development of off-odors, ropiness and/or discoloration at, or soon after, pack opening in our laboratory.

### 2.5. Isolation and Characterization of the Frankfurter Spoilage Microbiota During Storage

Potential compositional differences in the dominant spoilage LAB and other non-LAB contaminating biota between treatments were evaluated by the biochemical characterization of 240 representative isolates from the final frankfurter samples after 90 days of VP storage at 4 °C and 12 °C; when the products were either terminally spoiled or still acceptable for human consumption, depending on treatment. An equal number of 120 representative colonies were collected from each of the primary enumeration agar media for total bacteria (TSAYE; pH 7.3 ± 0.2) and LAB (MRS agar; pH 5.7 ± 0.1), respectively. Collection of the 120 isolates was done by picking ten colonies from one highest dilution TSAYE or MRS agar plate of one VP frankfurter sample for each treatment and experimental trial. Although the colony selection process was random, attention was paid to include all macroscopically different colony types, as analogously as possible to their visual occurrence, within the 10 isolates picked from each agar plate. By the above isolation procedure, 120 colonies (60/trial) were collected from each storage temperature, or otherwise, 80 colonies in total (40/trial; 20 from each temperature) were collected from each of the CN, SL and SL + SDA frankfurter treatments.

All isolates were grown in MRS broth at 25 °C, checked for purity by streaking on MRS agar, stored in MRS broth with 20% glycerol (Merck) at −30 °C. Fresh MRS agar cultures of all isolates were tested for Gram reaction by the 3% KOH method and catalase formation by dropping a 3% H_2_O_2_ solution (Merck) onto the cell biomass [37]. Only the gram-positive and catalase-negative, presumptively LAB, isolates were identified, based on few biochemical tests sufficient to assign them in different LAB groups of species, according to previous meat identification studies by Samelis et al. [37,39,40,41,42]. Briefly, all isolates were tested for cell morphology, production of carbon dioxide from glucose, ammonia from arginine hydrolysis, slime (dextran) from sucrose, growth in MRS broth at 4 °C, 37 °C and 45 °C, and fermentation of (Sigma-Aldrich Chemie GMbH, Steinheim, Germany) 13 basic (key) differentiating sugars: L-arabinose, cellobiose, galactose, lactose, maltose, mannitol, melibiose, raffinose, ribose, sucrose, sorbitol, trehalose and xylose, according to the procedures described by Samelis et al. [41].

### 2.6. Statistical Analysis

Two frankfurter production runs (independent experimental trials) were performed by analyzing two individual VP frankfurter samples on each sampling day per trial (*n* = 4). The microbiological data were converted to log_10_ units and along with the pH data were subjected to one-way analysis of variance using the Statgraphics Plus for Windows v. 5.2 (1995, Manugistics, Inc, Rockville, MD, USA) software. Means were separated by the Least Significance Difference (LSD) test at the 95% confidence level (*p* < 0.05) for determining differences in each frankfurter treatment with storage time. The same software was then used to find significant frankfurter treatment (CN, SL, SL + SDA) effects in correlation with the storage temperature effects and LAB growth-enumeration media (TSAYE and MRS) effects on each sampling day.

## 3. Results and Discussion

### 3.1. Physicochemical and Sensory Characteristics and Microbiological Quality of Fresh Frankfurters

The mean moisture, fat, protein and salt contents of the frankfurters after peeling (day-0 of storage) were 59.4 ± 1.0 %, 17.9 ± 0.9%, 11.4 ± 0.4% and 2.3 ± 0.2%, respectively, and were similar (*p* > 0.05) irrespective of organic acid salt treatment. Conversely, significant differences were noted in the initial pH between the SL + SDA and the CN and SL frankfurter treatments (Table 1). In specific, SDA at 0.25% in the formulation reduced the initial pH of the SL + SDA frankfurters by 0.3 to 0.5 units (*p* < 0.05) compared to the other two treatments, in agreement with the pH drop of similar pilot-plant frankfurter trials produced in the US previously [8].

Sensory evaluation by the industrial panelists found minor differences between the CN and SL frankfurter treatments; the SL + SDA treatment did not differ from the other treatments in appearance and color either. However, a partial loss of product elasticity, an acidic smell and a “vinegar-like” taste was reported for the frankfurters with 0.25% SDA by all panelists. Most of them described a slight feeling of ‘stiff side mouth’ and “throat burning” post-eating. These defects were not perceived by the CN or the SL frankfurters. A lower preference by panelists of beef frankfurters manufactured with 2% SL and 0.25% SDA, compared to other frankfurter treatments without SDA, were also reported by Morey et al. [12], who suggested studying changes in the SL/SDA treatment formulation in order to improve product quality.

No significant differences (*p* > 0.05) in the populations (log CFU/g) of total mesophilic bacteria (3.1 ± 0.4), LAB (2.6 ± 0.7), staphylococci (2.4 ± 0.4), enterobacteria (1.7 ± 0.4) and aerobic gram-negative bacteria (2.3 ± 0.3) were noted between frankfurter treatments before storage. Additionally, all frankfurter samples harbored less than 2.0 log CFU/g of enterococci, yeasts, and pathogenic staphylococci. Sulfite-reducing clostridia were absent/g (data not shown).

### 3.2. Effect of Organic Acid Salts on Microbial Growth in VP Frankfurters during Storage at 4 °C and 12 °C

LAB was the only microbial group that promoted major (*p* < 0.05) progressive growth in all VP frankfurter samples, and hence, sooner or later depending on the acid salt treatment, exceeded 6 log CFU/g during storage at 4 °C or 12 °C (Figure 2 and Figure 3). Within each treatment, the LAB populations grown on MRS agar at 25 °C (Figure 3) were similar (*p* > 0.05) to the total bacterial populations grown on TSAYE at 30 °C (Figure 2) during storage at both temperatures. This result indicated: (i) the TSAYE populations consisted mainly of LAB, and (ii) non-aciduric LAB species unable to grow on MRS agar, pH 5.7 ± 0.1, because of their sensitivity to acetate (i.e., the MRS formula contains 0.5% SA) were absent or present at low levels.

The populations of all other bacteria and yeasts remained below 3 log CFU/g, except of enterococci which exceeded 3 log CFU/g, but never reached 5 log CFU/g, in few spontaneous frankfurter samples stored at 12 °C (data not shown).

Growth of LAB was faster (*p* < 0.05) in all frankfurter samples stored at 12 °C than 4 °C, irrespective of acid salt treatment and enumeration agar medium (Figure 2 and Figure 3). However, the increases of LAB populations on MRS agar during refrigerated (4 °C) storage were strongly dependent on treatment (*p* < 0.05) since they were greater in the order: CN > SL > SL + SDA (Figure 3). In specific, compared to the CN and SL treatments, the SL + SDA treatment delayed (*p* < 0.05) growth of LAB by an average of 5.1 and 3.1 log units, and 3.0 and 2.0 log units, respectively, after 30 and 60 days of storage at 4 °C (Figure 3). After 90 days at 4 °C, the LAB populations still remained below the 8-log unit level in the frankfurters formulated with SL + SDA and SL (*p* < 0.05), whereas the respective LAB populations in CN frankfurters were more than 10-fold higher (Figure 3). Similar treatment-dependent growth patterns of the spoilage LAB were noted on TSAYE as regards the frankfurter samples stored at 4 °C (Figure 2).

The LAB populations also increased in the treatment order CN > SL > SL + SDA in all VP frankfurter counterpart samples stored at 12 °C (Figure 2 and Figure 3). However, retarding effects of both organic acid salt treatments on LAB growth were much weaker at 12 °C (*p* < 0.05) than at 4 °C, while again were similar on MRS (Figure 3) and TSAYE (Figure 2). Accordingly, LAB growth was faster under thermal abuse conditions, and thereby, the differences between treatments as regards the total LAB population levels were minor from day 30 to 90 of storage at 12 °C (Figure 2 and Figure 3). However, earlier samplings showed that the SL + SDA mixture was again the most effective treatment in delaying total LAB growth (on MRS agar) in the VP frankfurters by an average of 2.9 and 1.0 log units, and 2.0 and 1.2 log units, after 7 and 15 days of storage at 12 °C, respectively, compared to the CN and SL frankfurter treatments (Figure 3). Similar treatment-dependent early LAB growth retardations in the frankfurters stored at 12 °C were noted on TSAYE (Figure 2).

### 3.3. Effect of Organic Acid Salts on the pH of Spoiling Frankfurters during VP Storage at 4 °C and 12 °C

The pH decreased progressively during storage of all frankfurter samples stored at 12 °C and of the CN and SL samples stored at 4 °C (*p* < 0.05); in contrast, no pH decreases (*p* > 0.05) were noted in the SL + SDA samples stored at 4 °C (Table 1). Overall, the pattern of pH drop reflected the temperature-dependent growth pattern of LAB in each frankfurter treatment (Figure 2 and Figure 3). Clearly, the SL + SDA formula was the most effective treatment in preventing or significantly delaying the frankfurter pH reduction during storage at 4 °C and 12 °C (Table 1).

In-package gas and slime formation and lactic acid-fermenting off-odors at opening of the frankfurter packages decreased in the order SL + SDA (acceptable on day 90) < SL (mostly spoiled on day 60; always spoiled on day 90) < CN (always spoiled on day 60, or earlier) at 4 °C. Sensory defects developed much faster and were stronger during frankfurter storage at 12 °C. However, package swelling and excessive in-package ropy slime developed from day 15 to 30 in the CN and SL treatments only. In contrast, the SL + SDA sample packages stored at 12 °C either became ‘loose’ after the firmness of the packaging film with the frankfurter links was lost (exp. trial 1) or remained firm without visible spoilage defects throughout storage at 12 °C (exp. trial 2). Moreover, the SL + SDA frankfurter samples of the experimental trial 1 started accumulating small amounts of an in-package sticky slime rather than a watery ropy slime formed in the spoiled CN and SL samples after day 30. Conversely, the SL + SDA frankfurter samples of the experimental trial 2 were characterized by an unpleasant malty smell only after 30 days at 12 °C. Overall, the differences in pH drop (Table 1), total LAB growth patterns (Figure 2 and Figure 3) and the spoilage manifestation times suggested major compositional differences in the LAB species associations between the CN, SL and SL + SDA frankfurter treatments.

### 3.4. Characterization of the Spoilage LAB Isolates from VP Frankfurters Stored at 4 °C and 12 °C

All 240 isolates were LAB grouped and identified at the species level, according to their biochemical profiles presented in Table 2. No LAB isolate grew at 45 °C; all grew at 37 °C except of 31 isolates in group F. Only the sporadic isolates in groups C and D failed to grow at 4 °C; thus 96.3% of the isolates were psychrotrophic LAB. Most of them (63.4%; 153 isolates in total) were members of the facultative heterofermentative *Lb. sakei/curvatus* group (Table 2 and Table 3), which currently represents a new separate genus, *Latilactobacillus*, following the recent taxonomic proposal to reclassify the large and diverse genus *Lactobacillus* into 25 (23 new) genera [43]. All 153 *Lb. sakei/curvatus* isolates fermented ribose, but none fermented mannitol, raffinose, sorbitol and xylose (Table 2), four of the key-negative sugars for this cluster [44,45]. They were differentiated further into two major phenotypic groups, A and B (Table 2 and Table 3). The largest group B (35.4%; 85 isolates) included all typical *Lb. curvatus* biotypes that were unable to produce ammonia from arginine and ferment melibiose (Table 2). Group A (28.0%; 67 isolates in total) was more heterogeneous, and therefore, it was split into three subgroups: one major A1 to include all typical (arg+, mel+) *Lb. sakei* biotypes and two minor, A2 (arg+, mel-) and A3 (arg-, mel+), to include the atypical isolates of two intermediate *Lb. sakei/curvatus* biotypes, respectively (Table 2). Atypical biotype-A2 isolates were previously confirmed to be *Lb. sakei* by genotyping, including autochthonous strains from traditional, naturally fermented Greek dry salami originally isolated and biotyped by Samelis et al. [41,46] and later genotyped by Chaillou et al. [47]. Atypical isolates of the former “*Lb. curvatus* subsp. *melibiosus*” (arg-, mel+) biotype-A3 [44] also occurred naturally in traditional Greek taverna sausage [35]. To date, *Latilactobacillus (Lb.) curvatus* is clearly distinct at the genomic species level from *Latilactobacillus (Lb.) sakei* [43] after the invalid subspecies name “*Lb. curvatus* subsp. *melibiosus*” was rejected as a later synonym of *Lb. sakei* subsp. *carnosus* [48]. Discrimination of *Lb. sakei* subsp. *carnosus* from *Lb. sakei* subsp. *sakei* is not possible by biochemical criteria and thus requires molecular identification tools [45,49]. Actually the two *Lb. sakei* subspecies are biochemically intermixed [44] within the existing three distinct evolutionary lineages of this complex meat-specific LAB species [47]. Moreover, genotyping often provides controversial data to the biotype diversity of *Lb. sakei* [49]. The high intra-species phenotypic heterogeneity of previous *Lb. sakei* and *Lb. curvatus* isolates from Greek cooked or fermented meat products, particularly as regards their variable fermentation reactions with L-arabinose, cellobiose, lactose, maltose, sucrose and trehalose [35,37,40,41], was confirmed (Table 2). Therefore, for the purposes of this technological frankfurter study, we avoided splitting further subgroups A1, A2 and A3 of *Lb. sakei* and group B of *Lb. curvatus* into several additional biotypes of high phenotypic homogeneity without genotypic justification.

The second most frequent type of spoilage LAB (32.8%; 79 frankfurter isolates in total) was the obligatory heterofermentative, arginine-negative *Leuconostoc* or *Weissella* group of strains (Table 3). All of them had elongated coccoid cells of different sizes (Table 2). Most of them were strong gas- and slime-producing *Leuconostoc mesenteroides* subsp. *mesenteroides* isolates (group E), followed by weaker gas-producing *Leuconostoc carnosum* (group F) isolates which were intermixed as regards their ability to form slime from sucrose. The remaining gas-producing isolates were slime-negative *Lc. pseudomenteroides* or *Weissella paramesenteroides* (group G), two species that also are biochemically intermixed [23] and thus difficult to be distinguished by phenotypic criteria only. Genus and/or species specific genotypic id-tools are required to confirm identify of the leuconostoc-like isolates in groups F and G [42,50].

Altogether the isolates of *Lb. sakei/curvatus* and *Leuconostoc*/arginine-negative *Weissella* groups comprised the vast majority (96.2%) of the frankfurter isolates at spoilage (Table 3). The remaining nine isolates (3.8%) were sporadic LAB, which formed two minor facultative heterofermentative *Lactobacillus* groups, C and D (Table 2 and Table 3). Group D comprised typical arginine-negative strains of the *Lb. plantarum/paraplantarum* group which fermented all sugars tested except D-xylose (Table 2), currently assigned to the new genus *Lactiplantibacillus* [43]. Group C comprised arginine-positive lactobacillic isolates which, in addition to melibiose, fermented raffinose and xylose, but did not ferment mannitol and sorbitol (Table 2). They were neither *Carnobacterium* nor *Lb. algidus, Lb. fuchuensis* or *Lb. oligofermentans*, all associated with cold-stored VP or MAP meat or poultry products [22,45]. Additional biochemical tests plus genotyping are required to identify the sporadic group C isolates accurately.

### 3.5. Effects of the Storage Temperature on the Spoilage LAB Association of VP Frankfurters

The storage temperature had minor effects on the distribution of the predominant *Lb. curvatus* (group B) and *Lb. sakei* (subgroup A1) isolates, as well as, of the subdominant *W. paramesnteroides* (group G) isolates, irrespective of organic acid salt treatment (Table 3). In contrast, the isolation frequency of *Lc. carnosum* (group F) was ca. 5-fold higher from the frankfurter samples stored at 4 °C than 12 °C, whereas the isolation frequency of *Lc. mesenteroides* (group E) was ca. 4-fold higher from the frankfurter samples stored at 12 °C than 4 °C (Table 3). Similar temperature-dependent reversal distribution trends were also noted for the minor *Lactobacillus* groups. While all *Lb. sakei/curvatus* in subgroup A2 were isolated from refrigerated (4 °C) frankfurter samples, all except one of the atypical ”*Lb. curvatus/melibiosus”* isolates (subgroup A3) were isolated from thermally abused (12 °C) frankfurter samples. No *Lb. plantarum* (group D) or *Lactobacillus* sp. (group C) isolate was found at 4 °C (Table 3).

The natural prevalence of psychrotrophic strains of *Lb. sakei*, particularly *Lb. sakei* subsp. *carnosus*, and *Lc. carnosum* in refrigerated fresh and cured meat products is well known [1,2,6,25,26,27,28,29,30,34]. Compared to *Lb. sakei*, the natural prevalence of *Lb. curvatus* is lower in fresh refrigerated meats, but it increases in traditional meat fermentations [41,51] and cooked meat products containing fermentable sugars. Nevertheless, numerous *Lb. curvatus* strains promote abundant growth, similarly to *Lb. sakei*, at 4 °C [41,44], confirmed in this study (Table 2 and Table 3). Both species ferment strongly glucose and galactose (hexoses) and ribose (pentoses) added as commercial carbohydrate mixtures in raw or cooked cured meat formulations. However, when glucose and other fermentable sugars are deficient or depleted, as it happens in cold-stored fresh meats or processed meat products without added sugars, *Lb. sakei* has a strong competitive growth advantage over *Lb. curvatus* thanks to activation of secondary pathways, mainly the depletion of arginine [52], which is not hydrolyzed by *Lb. curvatus* (Table 2).

*Lc. mesenteroides* has also been reported as a primary post-process contaminating LAB species, often responsible for the rapid and offensive spoilage manifested as excessive VP or MAP blowing and in-package ropy slime formation, of a great variety of cooked cured meat products, including frankfurters, particularly under elevated temperatures (above 6 to 8 °C or mainly above 10 °C) of storage [1,3,32,37,39,40,53,54].

Unlike the storage temperature, the two primary growth media (TSAYE vs. MRS) did not affect the isolation frequency of each LAB group of species (Table 2).

### 3.6. Selective Pressure Effects of SL and SL + SDA on the Spoilage LAB Association of VP Frankfurters Stored at 4 °C and 12 °C

Major differential selective pressure effects of SL and SDA on the LAB species prevailing in frankfurter treatments were noted at both storage temperatures (Table 3, Table 4 and Table 5). Overall, the predominance of the *Lb. sakei/curvatus* group of isolates was strongly enhanced (87.5% in Table 3) by addition of 0.25% SDA in the formulation. Simultaneously, the SL + SDA treatment was less favorable for the prevalence of *Leuconostoc* spp. In specific, the isolation frequency of *Lc. mesenteroides* (group E) from the SL + SDA frankfurters was approximately 5-fold and 3-fold lower compared to the SL and CN frankfurters. Additionally, the isolation frequency of *Lc. carnosum* (group F) from the SL + SDA frankfurters was 5.5-fold and 3.5-fold lower compared to its isolation frequencies from the SL and CN frankfurters, respectively (Table 3).

Additional studies are required to elucidate why the inclusion of 1.8% SL singly in the frankfurter formulation favoured, or at least allowed, selective prevalent growth increases of *Lc. carnosum* at 4 °C and of *Lc. mesenteroides* at 12 °C. Probably the natural selection of *Lc. carnosum* strains was favoured under refrigeration despite their overall slower growth rates [4,36] underneath the dominant *Lb. sakei* or *Lb. curvatus* contaminants in cooked meats at 4 °C. A recent gene-based transcription study reported that three of the commonest meat spoilage psychrotrophic LAB species in Finland, *Lc. gelidum, Lb. oligofermentans* and *Lactococcus piscium*, display different survival strategies in response to competition, probably because their in situ evolution would rely on different metabolic pathways and growth substrates [55]. In this study, the high prevalence of psychrotrophic *Lc. carnosum* strains in the SL frankfurter samples at 4 °C was reversed in the SL samples at 12 °C in favor of *Lc. mesenteroides* strains, which dominated in both trials, particularly on TSAYE (Table 4 and Table 5). No *Lc. carnosum* isolates were recovered from the MRS agar plates of all samples stored at 12 °C (Table 5).

To summarize, the presence of SDA was a kind of prerequisite for reducing the in situ prevalence of *Lc. carnosum* and *Lc. mesenteroides* in the SL + SDA frankfurter samples of both experimental trials after 90 days of storage at 4 °C and 12 °C, respectively. At earlier storage times, SDA at 0.25% caused significant delays in the total LAB growth, particularly at 4 °C (Figure 2 and Figure 3). Hence, the retail shelf life of the SL + SDA frankfurter samples was much extended under refrigeration because the evolution of leuconostocs was suppressed in favor of *Lb. sakei/curvatus* strains. In specific, based on the storage times spoilage defects manifested themselves, the actual shelf life increment of the SL + SDA frankfurters was ca. 30 and 60 days compared to the SL-frankfurters and the CN-frankfurters, respectively, at 4 °C under vacuum. Interestingly, while typical *Lb. sakei* isolates (subgroup A1) were predominant, or at least numerous, in the SL + SDA frankfurter samples of the exp. trial 1, *Lb. curvatus* isolates (group B) dominated over *Lb. sakei* in the corresponding samples of the exp. trial 2 (Table 4 and Table 5). Particularly on MRS, *Lb. curvatus* comprised 90% of the LAB isolates from the SL + SDA frankfurters of the exp. trial 2 at both storage temperatures (Table 5).

The above differences in the spatial distribution of *Lb. sakei* and *Lb. curvatus* between the two individual frankfurter production runs (trials) may reflect seasonal variations in the LAB species composition of the natural contaminating biota persisting in the industrial meat plant environment. Another variation may be due to the usage of high-pH pork which is preferred technologically for the manufacture of emulsion-type sausages because of its higher water holding capacity compared to normal-pH pork. A recent microbial study by Charmpi et al. [51] noted that *Lb. curvatus* was more manifest in high-pH pork. Accordingly, the natural contamination and selective prevalence of *Lb. curvatus* may readily increase in frankfurter processing lines or under a constantly high pH (>6.0) on frankfurters formulated with SDA during refrigerated (mainly) or thermal abuse conditions of storage (Table 1 and Table 3).

As regards the in-package sticky slime that occurred in the SL + SDA frankfurters of the first trial only, where *Lb. sakei* isolates prevailed, this defect was probably associated with the ability of one or more prevalent *Lb. sakei* biotypes to produce slime in situ on the product surface; whereas the predominant *Lb. curvatus* biotypes in the SL + SDA frankfurters of the second trial might be unable for in situ slime formation. Nonetheless, all *Lb. sakei* and *Lb. curvatus* isolates of this study did not produce slime from sucrose in vitro (Table 2), unlike numerous previous isolates of *Lb. sakei* (mainly) and *Lb. curvatus* from Greek frankfurters and other cured cooked or fermented meat products [35,37,40,41]. Ropiness is a serious and frequent spoilage defect in cooked cured meat products [53,54], most recently reviewed by Iulietto et al. [6]. Overall, variations exist in the structure and viscosity of the extracellular polysaccharides formed by different LAB species or strains, many of which do not require the presence of sucrose in the (cured) meat substrate in order to form slime [6,53]. Therefore, additional meat product-specific investigations are required to characterize the type/s of slime, the reason/s for the occasional slime formation and the responsible LAB biotypes in frankfurters and other Greek meat products stored under VP or MAP conditions.

### 3.7. Practical Technological Aspects for Frankfurters and Other Cured Cooked Meats

From a practical industrial point of view, the present findings confirm and may further extrapolate previous data regarding major variations in the spoilage LAB species prevailing in different Greek cooked meat products [35,36,37,39,40,42]. Frankfurters belong to the smoked emulsion-type cooked meat product category that much differs from cooked ham and turkey [1]. However, the high final prevalence of gas- and slime-producing *Lc. mesenteroides* and *Lc. carnosum* in the SL frankfurters of this study (Table 3) corroborates strong ‘blown-pack’ spoilage defects caused by *Lc. mesenteroides* in retail SL-containing sliced VP cooked ham and turkey breast products of another industrial Greek plant during storage at 4 °C and 12 °C [37,40]. In contrast, non-slime-forming *Lb. sakei* biotypes similar to those in Table 2 predominated during storage at 4 °C and 10 °C in air, vacuum or 100% CO_2_ of Greek taverna sausages with an acetate content of 0.1% post-cooking due to the addition of SA [35]. Hence, acetate is amongst the most critical processing factors that enhance the natural selection and domination of *Lb. sakei* and *Lb. curvatus* over *Lc. mesenteroides* and *Lc. carnosum* and, to a lesser extent, *W. paramesenteroides* and *W*. *viridescens*, in comminuted smoked Greek cured meat products [36,39].

The results of this industrial frankfurter study are in general agreement with the findings of Benson et al. [20], who first reported that treatment of a model fresh (raw) pork sausage with a commercial SL/SDA blend altered the microbial dynamics dramatically, yielding a monophasic growth curve of a single species, *Lactobacillus graminis*, followed by a uniform selective die-off of most other very diverse LAB and non-LAB species in the uncooked meat population. Notably, *Lb. graminis* belongs to the *Lb. sakei/curvatus* genomic group of species, renamed as *Latilactobacillus graminis* comb. nov. [43]. Strains of *Lb. curvatus/graminis* also were more abundant than other LAB species in French fresh (raw) pork sausage manufactured with PL and SA, whereas the removal of lactate and acetate from the product formulation resulted in higher numbers of non-LAB spoilers, mainly *Brochothrix* and *Pseudomonas* [21].

Potential selective effects to the benefit of predominant lactobacilli, and likely associated with addition of PL (E326; declared on the product label) in the formulation of Belgian sliced cooked meat products, were recently noted by Geeraerts et al. [29,30]. In specific, high levels of lactobacilli were found in one sliced Belgian product at expiration date, which requires further validations [29]. Overall, numerous interactive factors may act differently, and hence, alter variably the spoilage LAB dynamics in cooked meat products from different countries, including the cold climate that favors psychtrotrophic *Lb. sakei* subsp. *carnosus*, *Lc. carnosum, Lc. gelidum* and carnobacteria strains to prevail in Northern European meat products and plants [5,25,26,27,28,29,30]. Nevertheless, the product type and the packaging and storage conditions remain the most decisive factors for the ultimate spoilage LAB species selection [39,40,56].

Finally, the strong effects exerted by SDA on delaying LAB growth and selecting for *Lb. sakei/curvatus* in the industrial frankfurters should be considered when applying commercial meat bioprotective LAB cultures which, apart from *Lb. sakei* strains, may contain acetate-sensitive *Carnobacterium* spp. strains [57]. Based on this study, the in situ performance of these mixed protective cultures may be altered, reduced or even suppressed after a potential failure of their *Carnobacterium* fraction to grow in cooked meat products formulated with SDA. Many studies have so far contributed to the development and application of effective cultures for meat biopreservation [3,32,57,58,59]. However, very few studies have considered comparing the in situ antilisterial activity of commercial protective LAB cultures in cooked meat products formulated with and without lactate/acetate blends [11], while no relevant studies have so far evaluated potential differential selective pressure effects of SDA or SL against the LAB strain constituents of commercial bioprotective mixed culture preparations in situ.

## 4. Conclusions

In conclusion, 0.25% SDA combined with 1.8% pure SL (3% of commercial SL product) extended the retail shelf life of industrial VP frankfurters by two months under refrigeration, compared to the standard frankfurters manufactured without SL or SDA. Hence, effective management of the spoilage LAB species communities in frankfurters and other processed meat and poultry products may be achieved through selective pressure effects exerted by lactate (SL, PL) and acetate (SA, SDA) combinations at optimal concentrations. Based on this study, further research is required (i) for optimizing the inclusion of SDA at levels lower than 0.25% to lessen texture and odor/taste defects sensed in the present SL + SDA frankfurters, too; (ii) for checking and optimizing the efficacy of existing commercial bioprotective cultures and for designing new bioprotective cultures for the meat processing industry. A feasible option that seems promising is to apply antagonistic *Lb. sakei/curvatus* strains with the mildest possible in situ spoilage potential, unable to cause ropiness, on frankfurters and other cooked meat products formulated with the most effective lactate-diacetate blends. This intelligent biotechnological approach would effectively combine natural preservative (abiotic) factors in the product core with novel bioprotective LAB culture/s on the product surface, relying on their strong in situ synergistic competitive exclusion effects to increase shelf-life and safety of frankfurters and other processed meat products.

## Figures and Tables

**Figure 1 foods-10-00074-f001:**
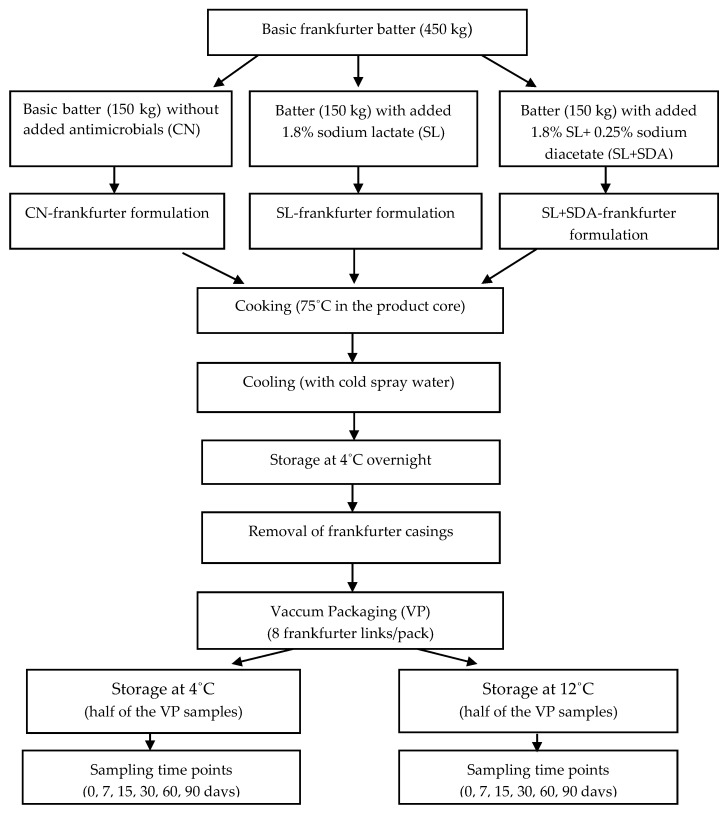
Preparation, formulation, storage and sampling of frankfurter treatments.

**Figure 2 foods-10-00074-f002:**
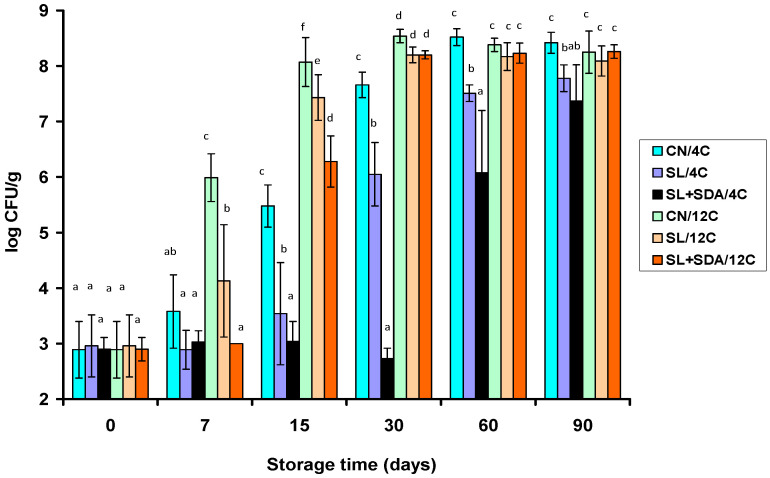
Mean total spoilage bacterial (TSAYE/30 °C) populations (*n* = 4) grown on vacuum-packaged frankfurters during storage at 4 °C and 12 °C. Frankfurter treatments contained 1.8% sodium lactate (SL), or 1.8% SL and 0.25% sodium diacetate (SL + SDA). Frankfurters without preservatives served as controls (CN). Whiskers for each point value in the graph indicate standard deviations. Mean value bars within each sampling time interval bearing different lowercase letters are significantly different (*p* < 0.05).

**Figure 3 foods-10-00074-f003:**
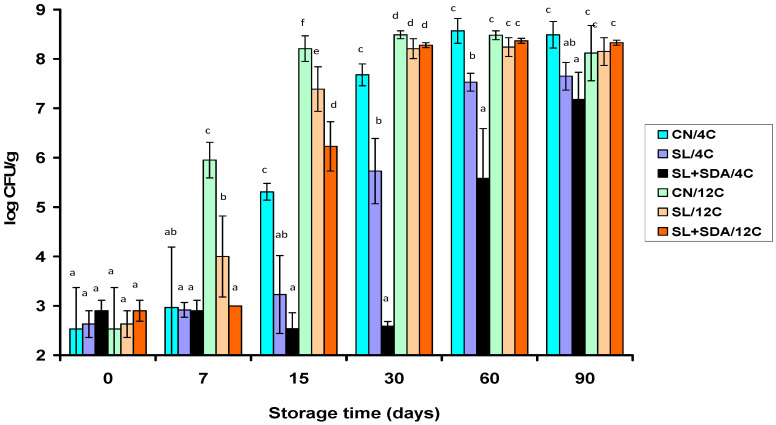
Mean total spoilage lactic acid bacteria (MRS agar/25 °C) populations (*n* = 4) grown on vacuum-packaged frankfurters during storage at 4 °C and 12 °C. Frankfurter treatments contained 1.8% sodium lactate (SL), or 1.8% SL and 0.25% sodium diacetate (SL + SDA). Frankfurters without preservatives served as controls (CN). Whiskers for each point value in the graph indicate standard deviations. Mean value bars within each sampling time interval bearing different lowercase letters are significantly different (*p* < 0.05).

**Table 1 foods-10-00074-t001:** Changes in pH of vacuum-packaged frankfurters during storage at 4 °C and 12 °C.

Storage Temperature	Frankfurter Treatment ^a^	pH Value of Frankfurter Samples During Storage (Days) ^b^					
		0	7	15	30	60	90
4 °C	CN	6.52 C ^b^ (0.05)	6.47 BC ^b^ (0.03)	6.46 BC ^b^ (0.03)	6.41 B ^c^ (0.03)	5.74 A ^bc^ (0.05)	5.67 A ^b^ (0.12)
	SL	6.40 B ^b^ (0.14)	6.39 B ^b^ (0.02)	6.39 B ^b^ (0.04)	6.44 B ^c^ (0.04)	6.32 B ^e^ (0.11)	6.15 A ^c^ (0.15)
	SL + SDA	6.05 A ^a^ (0.10)	6.12 AB ^a^ (0.16)	6.21 B ^a^ (0.05)	6.15 AB ^b^ (0.02)	6.17 AB ^d^ (0.06)	6.16 AB ^c^ (0.03)
12 °C	CN	6.52 E ^b^ (0.05)	6.42 DE ^b^ (0.08)	6.22 D ^a^ (0.09)	5.75 C ^a^ (0.22)	5.41 B ^a^ (0.12)	5.07 A ^a^ (0.24)
	SL	6.40 D ^b^ (0.14)	6.40 D ^b^ (0.02)	6.40 D ^b^ (0.03)	6.00 C ^b^ (0.04)	5.78 B ^c^ (0.03)	5.16 A ^a^ (0.27)
	SL + SDA	6.05 B ^a^ (0.10)	6.26 B ^a^ (0.19)	6.19 B ^a^ (0.09)	6.04 B ^b^ (0.11)	5.60 A ^b^ (0.17)	5.43 A ^b^ (0.02)

^a^ Frankfurters were formulated with 1.8% sodium lactate (SL) or with 1.8% sodium lactate and 0.25% sodium diacetate (SL + SDA), or without addition of the SL and SDA preservatives (control; CN). ^b^ Values are means (standard deviation) of two independent frankfurter production runs (experimental trials 1 and 2) with two individual VP samples analyzed for each frankfurter treatment and storage temperature per experimental trial (*n* = 4). Within a row, means with different uppercase letters are significantly different (*p* < 0.05). Within a column, means with different lowercase letters are significantly different (*p* < 0.05).

**Table 2 foods-10-00074-t002:** Biochemical identification of 240 LAB isolates from frankfurters formulated with or without organic acid salts, vacuum-packaged and stored at 4 °C or 12 °C.

LAB Species Identified	*Lactobacillus sakei*			*Lactobacillus curvatus*	Unidentified *lactobacillus*	*Lactobacillus plantarum*	*Leuconostoc mesenteroides*	*Leuconostoc carnosum*	*Weissella paramesenteroides*
**Group** **/subgroup-biotype**	**A1**	**A2**	**A3**	**B**	**C**	**D**	**E**	**F**	**G**
**Isolated from TSAYE**	**25**	**3**	**6**	**38**	**1**	**5**	**21**	**15**	**6**
**Isolated from MRS**	**26**	**4**	**3**	**47**	**2**	**1**	**16**	**16**	**5**
**No** **. of LAB isolates**	**51**	**7**	**9**	**85**	**3**	**6**	**37**	**31**	**11**
Cell morphology	R	R	R	R	R	R	CB	CB	CB
CO_2_ from glucose	-	-	-	-	-	-	+	+	+
ΝH_3_ from arginine	+	+	-	-	+	-	-	-	-
Growth at 37 °C	+	+	+	+	+	+	+	-	+
Slime from sucrose	-	-	-	-	+	-	++	10/31	-
Fermentation of									
L-arabinose	31/51	-	3/9	-	+	+	+	-	+
Cellobiose	30/51	-	+/+d	65/85	+	+	8/37	16/31	+/+d
Galactose	+	+	+	+	+	+	+	-	+
Lactose	22/51	-	3/9	49/85	+	+	+/+d	-	+/+d
Maltose	6/51	-	6/9	81/85	+	+	+	16/31	+
Mannitol	-	-	-	-	-	+	27/37	-	+/(+)d
Melibiose	+	-	+	-	+	+	+	-	+
Raffinose	-	-	-	-	+	+	+/+d	-	+/(+)d
Ribose	+	+	+	+	+	+	+	9/31	+
Sorbitol	-	-	-	-	-	+	NT	NT	NT
Sucrose	+	+	5/9	-	+	+	+	+	+
Trehalose	+	+	+	16/85	+	+	+	20/31	+
Xylose	-	-	-	-	+	-	+	-	8/11

+, positive reaction; -, negative reaction; +d, delayed positive reaction; (+) weak reaction; ++, strong positive reaction; 31/51, 31 out of 51 isolates in the group were positive; NT, not tested; R, rod-shaped cells; CB; elongated coccoid cells (coccobacilli), A1-A3, B-G, different group/subgroup-biotype.

**Table 3 foods-10-00074-t003:** Distribution of the identified LAB isolates in frankfurters formulated with or without organic acid salts, vacuum packaged and stored at 4 °C or 12 °C ^a^.

LAB Species/Biotype (Group in Table 2)	Number of Isolates (No)	Distribution of Isolates (% Total No)	Isolates from Frankfurter Samples at 4 °C ^b^	Isolates from Frankfurter samples at 12 °C ^b^	Relative Distribution of the Isolates within each Frankfurter Treatment ^b^		
					CN	SL	SL + SDA
*Lactobacillus sakei* (Group A/A1)	51	21.3	26 (21.7)	25 (20.8)	22 (27.5)	14 (17.5)	15 (18.7)
*Lactobacillus sakei/curvatus* (A2)	7	2.9	7 (5.8)	0 (0.0)	3 (3.8)	1 (1.2)	3 (3.8)
*Lactobacillus curvatus/’melibiosus’* (A3)	9	3.8	1 (0.8)	8 (6.7)	0 (0.0)	1 (1.2)	8 (10.0)
*Lactobacillus curvatus* (Group B)	85	35.4	48 (40.0)	37 (30.8)	20 (25.0)	21 (26.3)	44 (55.0)
***Lactobacillus sakei/curvatus* taxon**	**152**	**63.4**	**82 (68.3)**	**70 (58.3)**	**45 (56.3)**	**37 (46.2)**	**70 (87.5)**
Unidentified *Lactobacillus* (Group C)	3	1.3	0 (0.0)	3 (2.5)	1 (1.2)	2 (2.5)	0 (0.0)
*Lactobacillus plantarum* (Group D)	6	2.5	0 (0.0)	6 (5.0)	3 (3.8)	1 (1.2)	2 (2.5)
**Additional facultative heterofermentative *Lactobacillus* spp.**	**9**	**3.8**	**0 (0.0)**	**9 (7.5)**	**4 (5.0)**	**3 (3.7)**	**2 (2.5)**
*Leuconostoc mesenteroides* (Group E)	37	15.4	7 (5.8)	30 (25.0)	12 (15.0)	21 (26.3)	4 (5.0)
*Leuconostoc carnosum* (Group F)	31	12.9	26 (21.7)	5 (4.2)	11 (13.7)	17 (21.3)	3 (3.8)
*Weissella paramesenteroides* (Group G)	11	4.5	5 (4.2)	6 (5.0)	8 (10.0)	2 (2.5)	1 (1.2)
**Obligatory heterofermentative LAB**	**79**	**32.8**	**38 (31.7)**	**41 (34.2)**	**31 (38.7)**	**40 (50.1)**	**8 (10.0)**
**Total LAB isolates**	**240**	**100**	**120**	**120**	**80**	**80**	**80**

^a^ Treatments were: Frankfurters formulated with 1.8% sodium lactate (SL); Frankfurters formulated with 1.8% sodium lactate plus 0.25% sodium diacetate (SL + SDA); Control (standard) frankfurters without addition of the SL and SDA preservatives (CN).^b^ Numbers in parentheses show the percent relative distribution of each LAB species or biotype group in each frankfurter treatment or storage temperature.

**Table 4 foods-10-00074-t004:** Temperature-dependent and treatment-dependent LAB species distribution of 120 spoilage TSAYE isolates from two experimental trials of frankfurters manufactured with or without organic acid salts and stored at 4 °C and 12 °C in vacuum packages for 90 days ^a^.

LAB Species Identified	Storage at 4 °C						Storage at 12 °C					
(Group/Subgroup in Table 2)	CN		SL		SL + SDA		CN		SL		SL + SDA	
	Exp. 1	Exp. 2	Exp. 1	Exp. 2	Exp. 1	Exp. 2	Exp. 1	Exp. 2	Exp. 1	Exp. 2	Exp. 1	Exp. 2
*Lactobacillus sakei* (Group A/A1)	5	1	4	-	3	-	4	1	3	-	4	-
*Lactobacillus sakei/curvatus* (Group A/A2)	1	-	1	-	1	-	-	-	-	-	-	-
*Lactobacillus curvatus/ ‘melibiosus’* (Group A/A3)	-	-	-	-	1	-	-	-	-	1	1	3
*Lactobacillus curvatus* (Group B)	1	4	-	5	5	9	-	3	-	3	4	4
Unidentified *Lactobacillus* (Group C)	-	-	-	-	-	-	-	1	-	-	-	-
*Lactobacillus plantarum* (Group D)	-	-	-	-	-	-	1	2	-	1	-	1
*Leuconostoc mesenteroides* (Group E)	1	2	1	2	-	-	2	1	6	4	1	1
*Leuconostoc carnosum* (Group F)	2	2	4	2	-	-	1	2	-	1	-	1
*Weissella paramesenteroides* (Group G)	-	1	-	1	-	1	2	-	1	-	-	-

^a^ Treatments were: Frankfurters formulated with 1.8% sodium lactate (SL); Frankfurters formulated with 1.8% sodium lactate plus 0.25% sodium diacetate (SL + SDA); Control (standard) frankfurters without addition of the SL and SDA preservatives (CN).

**Table 5 foods-10-00074-t005:** Temperature-dependent and treatment-dependent LAB species distribution of 120 spoilage MRS isolates from two experimental trials of frankfurters manufactured with or without organic acid salts and stored at 4 °C and 12 °C in vacuum packages for 90 days ^a^.

LAB Species	Storage at 4 °C						Storage at 12 °C					
(Group/Subgroup in Table 2)	CN		SL		SL + SDA		CN		SL		SL + SDA	
	Exp. 1	Exp. 2	Exp. 1	Exp. 2	Exp. 1	Exp. 2	Exp. 1	Exp. 2	Exp. 1	Exp. 2	Exp. 1	Exp. 2
*Lactobacillus sakei* (Group A/A1)	4	3	2	1	3	-	4	-	4	-	5	-
*Lactobacillus sakei/curvatus* (Group A/A2)	2	-	-	-	2	-	-	-	-	-	-	-
*Lactobacillus curvatus/ ‘melibiosus’* (Group A/A3)	-	-	-	-	-	-	-	-	-	-	2	1
*Lactobacillus curvatus* (Group B)	-	4	2	5	4	9	3	5	2	4	-	9
Unidentified *Lactobacillus* (Group C)	-	-	-	-	-	-	-	-	-	2	-	-
*Lactobacillus plantarum* (Group D)	-	-	-	-	-	-	-	-	-	-	1	-
*Leuconostoc mesenteroides* (Group E)	-	1	-	-	-	-	2	3	4	4	2	-
*Leuconostoc carnosum* (Group F)	3	1	6	4	1	1	-	-	-	-	-	-
*Weissella paramesenteroides* (Group G)	1	1	-	-	-	-	1	2	-	-	-	-

^a^ Treatments were: Frankfurters formulated with 1.8% sodium lactate (SL); Frankfurters formulated with 1.8% sodium lactate plus 0.25% sodium diacetate (SL + SDA); Control (standard) frankfurters without addition of the SL and SDA preservatives (CN).

## Data Availability

The microbiological data presented in this study, as well as, representatives of the common meat LAB species or strain biotypes identified are available on request from the corresponding author.

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
