# Peer review of "Growth Inhibitory and Selective Pressure Effects of Sodium Diacetate on the Spoilage Microbiota of Frankfurters Stored at 4 °C and 12 °C in Vacuum"

_foods, 2021, doi:10.3390/foods10010074_

Round 1

Reviewer 1 Report

The study focuses on the use of SL and SDA in order to prolong the shelf life of commercial frankfurter. 

The research is interesting and it could be useful and transferable to the industries.

Authors used traditional microbiological techniques, which nowadays are often forgotten in favor of more attractive molecular methods, because they are considered innovative. In this case, and for the purpose of the work, I agree with the authors' choice, as often these methods (such as NGS) do not allow to accurately trace the identification of the species actually involved, but only allow approximate predictions, or the identification at a genera level. But the recognition of the species is fundamental to allow the understanding of the eventual spoilage. The work is very well written and the discussion is thorough. The materials and methods are clear and reproducible. I congratulate the authors for their well structured and described work.

The only observation that I believe needs to be taken into consideration, to increase the value of the work, is that there is no monitoring of pathogenic microorganisms, such as Salmonella and, in particular, Listeria monocytogenes. Whenever production protocols for increasing shel-life are evaluated, I believe it is also essential to evaluate the possibility of development of pathogenic microorganisms, because before being able to affirm that shel-life can be prolonged, it has to be sure that there is no development of pathogens. In this case, in the SL and SDA theses there is an increase in pH, which could favor the development of these microorganisms, especially Listeria, since it can survive and multiply even at refrigeration temperatures. Therefore I suggest the authors to verify if the proposed increase in shelf-life is compatible with the possible development of Listeria monocytogenes or Salmonella. Furthermore, from the data shown, it seems to me that the increase in shelf life can only be considered up to 30 days. I therefore suggest to the authors to effectively indicate in the discussion and in the conclusion the actual useful shelf life possible increment.

Reviewer 2 Report

Comments to foods-1028660

Title:
Sodium diacetate, combined with sodium lactate, extends the shelf life of vacuum-packaged frankfurters by delaying total LAB growth and reducing the prevalence of Leuconostoc carnosum and Leuconostoc mesenteroides strains with a high cured meat spoilage p

Authors: John Samelis * , Athanasia Kakouri

In this study, differential spoilage patterns were associated with a major reversal of the LAB biota from gas - and slime - producing Leuconostoc mesenteroides and Leuconostoc carnosum in the CN and SL frankfurters to Lactobacillus sakei/curvatus in the SL SDA frankfurters. Thus, SL SDA extends the retail shelf life of VP frankfurters by delaying total LAB growth and selecting for lactobacilli with a milder cured meat spoilage potential than leuconostocs, particularly under refrigeration.

Recommendations for Authors:

The present manuscript is very interesting however in the current state in addition to having typographical errors that should be corrected, the following comments should be taken into account

Line 2: The title needs to be improved. The phrase that is used as a title seems more a result than the title of the article.

Line 23: Explain the meaning of VP

Line 32: Explain the meaning of LAB

Line 87: It would be recommendable to make a graphic scheme illustrating the preparation of the samples, storage and sampling. This would help to better understand the experimental design

Line 105: It is necessary to include a space

Line 153: The sensory attributes studied and the methodology used have not been explained. What scale was used?

Line 154: How do you ensure that the characterization of the panelists is reproducible and consistent?

Line 156: It is not appropriate to assess the "overall acceptability" attribute in this study because the panel used is not a consumer´s panel. This parameter is measured when comprises consumer´s studies.  In this work the panel that evaluated is a group of 5 members who have experience and therefore they are considered to be a trained panel and they cannot represent the acceptability of the product

Line 190: You must explain in detail which treatment or statistical tool you have carried out. It is not explained which are the treatments or effects that have been studied

Line 208: you should improve the title of the table

Line 211: What statistical treatment did you carry out?

Line 215: There should be further clarification of the comparison of averages of values between rows and the same for averages between columns. In the current format it is confusing and not well differentiated

Line 208: The sensory attributes studied and the methodology used have not been explained. What scale was used?

Line 222: As I mentioned above, It is not appropriate to assess the "overall acceptability" attribute in this study because the panel used is not a consumer´s panel.

Section 3.2, 3.3, 3.4, 3.5, 3.6, you should summarize the title of the section and explain the most important aspects that are studied in this section

Line 267: The title of this figure should be summarized

Line 275: The title of this figure should be summarized

Table 2, 3, 4 : you should improve the title of the table

Line 454: The title of the table must begin with a capital letter

Line 545: separate words

Round 2

Reviewer 2 Report

No comments